# Vitamin Solutions Effects on Reproduction of Broodstock, Growth Performance, and Survival Rate of *Pangasius* Catfish Fingerlings

**DOI:** 10.3390/ani14152203

**Published:** 2024-07-29

**Authors:** Chau Thi Da, Bui Thi Kim Xuyen, Thi Kieu Oanh Nguyen, Van Tai Tang, Pham Thi Thu Ha, Minh Tan Pham, Håkan Berg

**Affiliations:** 1Faculty of Applied Sciences, Ton Duc Thang University, Ho Chi Minh City 70000, Vietnam; chauthida@tdtu.edu.vn (C.T.D.); phamminhtan@tdtu.edu.vn (M.T.P.); 2Viet Uc Pangasius Joint Stock Company, Vinh Hoa Commune, Tan Chau District, Long Xuyen City 90000, An Giang Province, Vietnam; btkxuyen@gmail.com; 3Mekolink Co., Ltd., My Hoa Ward, Long Xuyen City 90000, An Giang Province, Vietnam; ntkoanh.ts@gmail.com; 4Faculty of Technology, Dong Nai Technology University, Bien Hoa City 76100, Vietnam; tangvantai@dntu.edu.vn; 5Queensland Alliance for Agriculture and Food Innovation, The University of Queensland, St. Lucia, QLD 4072, Australia; phamthithuhabt@gmail.com; 6High Agricultural Technology Research Institute for Mekong Delta (HATRI), Can Tho City 94955, Vietnam; 7Department of Physical Geography, Stockholm University, 106 91 Stockholm, Sweden

**Keywords:** *Pangasius* catfish, broodstock nutrition, reproduction, growth performances, fingerling, survival rates

## Abstract

**Simple Summary:**

This study was carried out to examine how different diets with vitamin solutions/vitamin premixes and plant oils (algae and fungi oils) affect the reproduction of *Pangasius* catfish broodstock, growth performance, and survival rates of fish larvae and fingerling. The experiment was designed as a factorial setup, with *Pangasius* catfish broodstock fed six different diets in triplicate groups of ten fish (three males and seven females) cultured in separate hapa nets in an the earthen pond for about two months. Results showed significant differences among the six tested diets. Diets containing 35% CP, supplemented with 0.6% vitamin solutions and 1.26% algal oil, produced the best results in terms of the highest growth performance, reproductive indices, and fingerling survival rates. These findings provide valuable information for *Pangasius* catfish farmers and the fish production industries in the Mekong Delta, Vietnam.

**Abstract:**

This study evaluates the effect of different diets supplemented with vitamin solutions on *Pangasius* catfish broodstock reproduction, growth performances, and the survival rates of fish larvae and fingerling. The growth and reproductive performances of breeders fed with different test diets showed significant differences among the six tested diets (*p* < 0.05). The highest final body weight (FBW), weight gain (WG), daily weight gain (DWG), specific growth rate (SGR) of broodstock, and survival rate of *Pangasius* fingerlings were found in Treatment 5, which contained 0.6% H-OVN mixed with 12.6% algal oil, and Treatment 3, which contained 0.6% vitamin premix H-OVN. The average gonadosomatic index (GSI), relative fecundity index (RFI), fertilized eggs, hatching rates of eggs, and survival rate of fingerlings was 9.1 ± 2.8 (6.7–12.8%), 133,224 ± 39,090 (104,267–199,512 eggs/kg), 77.9 ± 22.2 (62.2–93.6%), and 45.3 ± 17.4 (22.0–66.3%), respectively. The findings of this study showed that the diet containing 35% CP contents supplemented with 0.6% vitamin premix H-OVN mixed with algal oils showed the highest results in terms of growth, reproductive performance indices, and survival rates of *Pangasius* catfish fingerlings.

## 1. Introduction

In Vietnam, *Pangasius* catfish is an important species for commercial aquaculture, providing high economic revenues, with an annual production of approximately 1.52 million tonnes [1]. It is one of the country’s most significant export products, contributing substantially to Vietnam’s economy. The Mekong Delta region, in particular, is a hub for *Pangasius* catfish farming, providing livelihoods for thousands of farmers and workers in the aquaculture sector. Currently, there are some major challenges faced by farmers in producing high-quality fingerlings, as inbreeding and lack of genetic diversity can lead to lower growth rates, poor disease resistance, and overall reduced performance of the fish. Farmers often face challenges in sourcing and affording quality feed for broodstock fish. Inadequate nutrition of brooder can lead to poor growth rates, higher mortality, and increased susceptibility to diseases of fingerling.

Recent reports indicate that low survival rates and unstable supply of fish fingerlings throughout the year may become critical bottlenecks for the continued growth of the *Pangasius* aquaculture sector in the Mekong Delta, Vietnam [2,3,4]. *Pangasius* broodstock spawn all year round, but they have low and variable survival rates for fish fry and fingerlings, rarely reaching over 20–25% during the main spawning season and only 12–15% after the main spawning season [2,4,5,6,7]. In the Mekong Delta, Vietnam, most *Pangasius* catfish hatcheries and nurseries feed their broodstock fish with the same commercial diet as used for the grow-out of commercial fish. This diet, containing 28–30% available crude protein (CP) and 3.5–5% lipid content, may not contain the appropriate nutrient levels for an optimized spawning success of brooders and improved quality of fish fry [4,5,8,9]. Earlier research showed that low-quality broodstock nutrition, feeding, and feed management provide major challenges for fish producers, alongside decreased genetic integrity, inbreeding of fish, and seed degeneration [3,4,7,10,11,12]. As indicated by several researchers, better quality broodstock diets, formulated to increase successful reproduction and better fish management practices, would allow producers not only to reduce the number of broodstock needed to meet eggs and fry production goals but also produce better quality fry and meet the fry demands of fish farmers’ year round [13,14,15,16].

Many studies are being carried out around the world investigating the human health benefits of fungi and algae [17,18,19]. Several species of algae and fungi, including lichenized fungi (lichens), have the ability to biosynthesize biologically active compounds and are potential sources of natural antibiotics and antioxidants that could be used as supplementary medicine and food sources for human and aquatic animals [20]. The main nutrients found in mushroom-fruiting bodies are proteins, carbohydrates, and fats, including essential fatty acids (EFAs), fiber, and vitamins and minerals. Algae biomass and lichens are a renewable source with many valuable active substances that have a wide range of applications in many industries, such as food, chemicals, agriculture (including animal and aquatic feeds), pharmaceuticals, cosmetics, and medicines [17,18,21]. Microalgae provide essential amino acids and valuable triglycerides, such as lipids, vitamins, and pigments, making them suitable as nutritional supplements in animal feed and aquafeed formulations [17,18,22,23]. It has also been reported that microalgae supplements in diets improve the fatty acid profile of farmed fish and shrimp by improving the ω-3/ω-6 ratio, increasing polyunsaturated fatty acid (PUFA) content, and enriching long-chain PUFAs [22,24]. Studies on the application of microalgae species rich in EPA or DHA in aquaculture include Pacific white shrimp, Giant tiger prawn, Giant freshwater prawn [22,25,26], Gibel carp [24], Tilapia [23,26], European seabass [27], and Common carp [26]. Several studies reported that microalgae oils have the potential to replace fish meal and fish oils in aquaculture and ensure sustainability standards. It can be used directly as a supplement source in animal feed and aquafeed feed formulations to improve reproduction and produce good quality fish eggs, fish fry, and yields [18,23,28].

In recent years, dietary protein, lipid (fats, fatty acids), vitamin, and energy requirements of many commercial catfish species (including Channel catfish, Black catfish, Bagrid catfish, and African catfish) have been widely examined [29,30,31,32,33,34,35,36,37,38,39,40]. However, studies focusing specifically on broodstock nutrition of *Pangasius* catfish species are still limited. This absence of research may be because it is generally considered to be of high cost, as it requires a long period of feeding broodstock fish before any effects can be seen on fish fecundity, egg quality, and hatching success [13,14,15]. Still, the quality of feed ingredients, feed quality, and feed utilization by broodstock fish species are key factors in improving the reproductive performance, egg and sperm quality, hatchability, and enhanced survival rate of fish species [13,14,15]. Currently, limited research is available on broodstock nutrition and its potential effects on reproductive performance, fish fry survival rates, and the quality of fingerlings and seed when vitamins and fatty acids from algae oil and fungi oil are added to the diet of *Pangasius* catfish broodstock. A study on *Pangasius* catfish in Vietnam recommended that a standard for the conditioning feeds of *Pangasius* catfish broodstock needs to be developed [41].

This feeding trial was conducted to evaluate the effects of different diets supplemented with vitamin solutions on growth performance, broodstock reproduction, hatchability, and the survival rates of fry and fingerlings of *Pangasius* catfish. The specific objectives of this study were aimed to assess the reproductive success of broodstock, growth performance, and survival rates of *Pangasius* catfish fed by different enriched diets supplemented with vitamin solutions and plant oils to provide practical recommendations for the *Pangasius* catfish farmers and aquaculture industry based on the findings of the study. The hypothesis tested in this study was that a diet containing approximately 35% crude protein (CP), supplemented with different vitamin solutions and plant oils (algal oil and fungal oil), can improve the reproductive performance and egg quality of broodstock, as well as enhance the survival rates of fry and fingerlings of *Pangasius* catfish. The findings of this research provided valuable information for *Pangasius* catfish farmers and the fish production industry in the Mekong Delta, Vietnam.

## 2. Materials and Methods

### 2.1. Study Site and Research Layout

The experiments were carried out at the *Pangasius* catfish broodstock farm and hatchery in My Thoi wards, Long Xuyen city, An Giang province, Vietnam.

Two experiments were carried out, one outdoors and one indoors. In the outdoor experiment, 180 *Pangasius* catfish brooders were fed six test diets supplemented with differing amounts of vitamin premixes, algal oil, and fungal oil in a series of 18 hapa nets in an earthen pond of 1500 m^2^, with three replicates for each diet. The experiment was conducted during October–February, which is after the main spawning season. The experiments of breeding, larvae rearing, and fingerling rearing were conducted in an indoor hatchery. These experiments aimed to evaluate the quality of eggs, egg hatchability, growth performances, fingerling production, and the survival rates of fingerlings of *Pangasius* catfish. The research layout of this study is presented in Figure 1.

### 2.2. Pond Preparation and Management of the Broodstock

The earthen pond used for the broodstock experiment was prepared by pumping out the water and treating it with 150 kg lime (CaCO_3_). After this, the pond was filled with new freshwater from a reservoir pond. A system of 18 hapa nets with a 4.0 mm mesh size was used to test six treatments in triplicates. Each hapa net was suspended and tied to four Melaleuca poles. The sides and bottoms of each hapa net were scrubbed and cleaned every two months, and at least 20–30% of the water was exchanged monthly during the cultivation of the broodstock fish.

### 2.3. Selection Criteria of Broodstock

The broodstock were obtained from a broodstock fish pond at the *Pangasius* catfish farm of NAVICO. A total of 180 brooders fish at the pre-maturation stage and 3–3.5 years old were selected from the broodstock fish pond and transferred to the 18 hapa net system (6 treatments in triplicates), where each hapa net was 3.5 m × 3.5 m × 3.0 m (length × width × depth) (Figure 1). At the beginning of the experiments, the body weight, length, and belly width of the broodstock fish were measured. The female breeders were checked and selected based on having a large soft belly, healthy external appearance, good agility condition, and uniform egg size. The male breeders were selected based on the thickness of the milt/semen obtained by hand stripping, healthy external appearance, good agility condition, and large size [42].

### 2.4. Experimental Diet Preparation and Feeding Practice

The experimental diets were formulated to meet the nutritional requirements for striped catfish broodstock, with approximately 35% CP supplemented with vitamin premix, algal, and fungal oils (Table 1). The experimental feed ingredient sources were soybean meal (45% CP) and Kien Giang fish meal (55% CP), poultry by-product meal (65% CP), and soybean oil, which were purchased from the local markets in An Giang and Dong Thap provinces. Fish oil (Tuna oil), choline chloride (50% choline), mineral premix for fish, vitamin premix algal oil, and fungal oil were provided by DSM SINGAPORE INDUSTRIAL PTE. LTD, Mapletree Business city, Singapore.

The detailed compositions of the vitamin premix, mineral premix, and algal oil are presented in Table 1. Two tonnes of floating pelleted feeds (5.0 mm diameters) of the six test diets of the experiment were produced at the Aquafeed production of Dong A plant, Cao Lanh city, Dong Thap province, Vietnam. The broodstock fish were reared and fed with the experimental diets for 65 days. Broodstock fish of each treatment were fed by hand to apparent satiety, at a rate of about 3–5% of body weight, twice a day at 8:00–9:00 a.m. and 4:00–5:00 p.m. The chemical composition of the test ingredients and diets are shown in Table 2 and Table 3.

### 2.5. Broodstock Experimental Design

The experiment was set up as a factorial design with *Pangasius* catfish broodstock fed six different diets in triplicate groups cultured in a hapa net system in an earthen pond (1500 m^2^) with a depth of about 2.5 m. Ten fishes with a mate ratio of 3 males/7 females were reared in each hapa net for about two months to acclimatize them to the conditions in the hapa net. The average initial weights of the broodfish of females and males used in the study were about 6.0 ± 0.7 (4.5–7.7) kg and 4.5 ± 0.6 (3.5–6.2) kg, respectively. Before the experiment began, all the fish were fed daily on the same commercial diet containing 24% CP. Feeding was carried by hand from a small boat between 4 and 5 p.m. each day, at a rate of 3–5% of body weight, until fish reached apparent satiety.

### 2.6. Induced Spawning and Larvae Rearing Practices

#### 2.6.1. Mature Broodstock Fish Selection for Induced Spawning

After feeding of the broodstock fish for 65 days with the experimental diets, the egg quality and development status of individual female fish were checked with the help of a catheter. Mature females were identified by their big, round, and soft bellies, along with reddish, swollen ventral genital pores. Male broodstock fish were identified by observation of their genital papilla, which oozed milt/semen when they were ready to breed, the presence of a slight stripe on the abdomen, and the quantity and quality of their sperm/milt, which were checked by stripping.

Mature male and female broodstock fish in good condition from each treatment were selected, marked, and quarantined for 1–2 days in separate rectangle tarpaulin tanks with 10 m^3^ of water for female breeders and 5 m^3^ of water for male breeders to acclimatize to the water environmental condition in the hatchery before the induced breeding procedure commenced.

#### 2.6.2. Induced Stripping of the Broodstock

The spawning of the mature broodstocks was induced in hatcheries using human chorionic gonadotropin (hCG) injections. The female fish were given 4 injections, each injection with a different dose of hCG: 200, 300, 700, and 2700 UI/kg, while male fish were injected only once, at the same time as the final injection of the females. The detailed protocol of hCG doses, time for fish injection, and injection dose calculation are given in the supplemental information. Females ovulated from 10–12 h after the last injection at a water temperature of 27–28 °C. Eight hours after injecting the last dose of hCG, female breeders were checked for eggs by slight stripping on the belly to ensure that the ripe eggs were in stage IV condition. Eggs and milt/semen of fish breeders from each treatment were then drily striped and slowly poured into each other in small plastic tubs, each stripping into a separate tub. Eggs and body weight of each female from the different treatments were weighed for their gonadosomatic index determination. Milt/semen and eggs of different treatments were stirred around for two minutes by using chicken feathers and then washed 2–3 times with clean water (Figure 1). A tannin solution with a 5% concentration was added to the tub to remove adhesiveness (stickiness). The mixture was slowly poured on the mixed eggs in a plastic tub of each treatment and stirred for 1–2 min, and then thoroughly rinsed with clean water 2–3 times. One gram of eggs of each treatment was sampled and measured in triplicate to evaluate the egg fecundity, egg number, and egg size, using a microscope (Carl Zeiss Microscopy, Germany) at 4× and 10× magnifications. The wet weight of eggs was determined using an electronic balance (Mettler Toledo, Swiss).

#### 2.6.3. Egg Incubation and Larvae Fish Nursing Practices

An incubator system was set up in a closed re-circulation system in a series of 18 hatching jars (Weiss-shaped incubators) for six treatments in triplicates with a water volume of about 18 L for each hatching jar (Figure 1). The water supply for the hatching jars of the incubator system was taken from a reservoir tank, treated with a biological filter and passed through an ozone generator before being supplied to each hatching jar. The water flow through the incubator system had been adjusted to moderate flow so the eggs were stirred and not allowed to settle at the bottom of the jars. Fertilized eggs were stocked at an average density of 44,340 (15,800–88,890) eggs per hatching jar. The fertilized eggs of each treatment were incubated in the hatching jar system for 35–36 h at a water temperature of 25–27 °C during December.

After hatching, all larvae fish were moved and delivered into 18 different nursing tanks (six treatments in triplicates) with 0.5 m^3^ of water in each tank and reared for 48 h. Aeration with two air-stone diffusers was provided to each tank via a moderate-pressure electrical blower. The total length and height of larvae after hatching were measured under a microscope (Carl Zeiss Microscopy, München, Germany), and the wet weight of hatchlings was measured using an electronic balance (Mettler Toledo, Urdorf, Swiss).

### 2.7. Larvae to Fingerlings Rearing Experiment

The larvae remained in the nursing tanks for the first 48 h until the yolk sacs had been absorbed and then transferred to larger indoor tanks for the next 15 days, from December 2020 to January 2021. There were 24 tarpaulin tanks placed indoors, each tank with a water volume of 1 m^3^ (1 m × 1 m × 1 m for each tank) and a stocking density of 15,000 larvae/tank. They were supplied with water, which had been treated with a biological filter. The experiments were covered by a blue tarpaulin and a green net for sun and rain shade to control the temperature during the winter season (Figure 1). The rearing tarpaulin tanks were also covered with a long white tarred canvas to maintain the water temperature at nighttime (Figure 1G). Two days after hatching and until the 5th day, the larvae fish were fed with Artemia (Vinh Chau Artemia products) at a rate of Artemia to larvae of 5:1. Between 6–15 days after hatching, the larvae fish were also fed a mixture made up of 50% brine shrimp + 50% of UV-milk powder feed with 42% CP content (Commercial feed powder (Name: Tomboy TB0). The larvae were fed 4 times daily to apparent satiety at 8.00 a.m., 11:00 a.m., 2:00 p.m., and 5:00 p.m. At the beginning of the experiment, a sample of 30 larval fish from the nursing tank from each treatment was weighed using a digital scale and measured using a microscope (Mettler Toledo, Swiss) for the evaluation of growth performance indices. All larvae fish from each experimental treatment were harvested, counted, and weighed at the end of the experiments to estimate the production and final survival rates of the fish.

The second experiment of fry-to-fingerling rearing was conducted after finishing and harvesting the fish larvae from the first experiment. In total, 2000 fry per tarpaulin tank were reared for 45 days using the same facility, equipment, and diet treatments (6 treatments in 4 replicates) as the first experiment of larvae rearing (Figure 1). Fish were fed 2 times per day with the experimental milled diets to apparent satiety at 8.00 a.m. and 5:00 p.m. At the beginning and at the end of the experiment, a sample of 30 fish in each nursing tank from each treatment was weighed and measured in the same way as in the first experiment. Fish of each treatment during the experiment were collected every fortnight to measure weight and length gains for the evaluation of growth performance indices. All fry of each experimental treatment were harvested, counted, and weighed at the end of the experiments to estimate fingerling production and final survival rates of fingerlings.

### 2.8. Water Quality Monitoring

Temperature (T °C), dissolved oxygen (DO mg/L), and pH in the earthen pond of the broodstock cultivation and the rearing tank system for fish larvae were recorded daily with a DO meter. Water samples for nitrogen (NO_2_^−^ mg/L) and ammoniac (NH_3_^+^) analyses were collected twice a month and kept cool in the refrigerator until they were analyzed using the Hach Lange cuvette test method (DR2800 visual spectrophotometer, Hach Lange Gmbh, Berlin, Germany).

### 2.9. Chemical Analysis

Samples of experimental test ingredients and diets were analyzed by using standard methods [43]. Dry matter was determined by drying in an oven at 105 °C for 24 h. Nitrogen (N) was determined by the Kjeldahl method, and crude protein (CP) was calculated as N × 6.25. Crude fat (EE) content was analyzed using the Soxhlet method after acid hydrolysis of the sample. Ash content was determined by incineration in a muffle furnace at 550 °C for 12 h. Amino acid profiles of ingredients and diets (Table 2 and Table 3) were analyzed by high-performance liquid chromatography according to [44]. The fatty acid composition of the diets (Table 3) was determined using the total lipid extracts of the diets that were transesterified with boron trifluoride. Laboratory analysis of feed ingredients and diets was conducted at the Advanced Laboratory, Department of Science, Can Tho University, Vietnam.

### 2.10. Calculation

The following calculations were made:Fertilization (%) = [Number of fertilized eggs/total number of eggs in the batch] × 100.
Hatching rate (%) = [Number of hatched eggs/total number of eggs in the batch] × 100.
Ripe eggs (%) = (Number of eggs with yolk position near one edge of the egg/total number of eggs counted) × 100.
Gonadosomatic index (GSI%) = (Gonad weight/total body weight) × 100.
Relative fecundity index (RFI) = (Total number of eggs in female ovary/total weight of female).
Weight gain = Final body weight − Initial body weight.
Length gain = Final body length − Initial body length.
Daily weight gain (DWG) = (W_f_ − W_i_)/T),
where W_f_ and W_i_ refer to the mean final weight and the mean initial weight, respectively, and T is the feeding trial period in days.
Specific growth rate (SGR%) = [(ln W_f_ − ln W_i_)/T] × 100.
Food conversion ratio (FCR) = [total feed intake (g)/total wet weight gain (g)].
Survival rate [(SR%) = (TF_f_/TF_i_) × 100],
where the TF_f_ is the total number of fish at the finish (harvest), and TF_i_ is the total number of fish at the start.

### 2.11. Statistical Analysis

All data on induced spawning, egg fecundity, early life stage development, hatching rate, growth performances of broodstock and larvae fish, survival rate of fry and fingerlings, and water quality parameters were statistically analyzed by using one-way ANOVA to check for overall significant differences between group means, if significant, followed by Tukey’s HSD test for detailed pairwise comparisons (*p* ≤ 0.05 level of significance), with the software of MINITAB Statistic program (version 16) applied.

## 3. Results

### 3.1. Chemical Composition and Essential Amino Acid Content of Feed Ingredients and Test Diets

The result of the nutrient contents of feed ingredients showed that nutrient contents are highest in poultry by-product meal, followed by fish meal, soybean meal, and wheat flour (Table 2). The chemical analysis composition of the six diets used for the *Pangasius* catfish broodstock cultivation is presented in Table 1. The crude protein (CP) contents of the experimental diets were approximately 350 g/kg DM. The highest crude fat contents were found in the diets of Treatment 5, Treatment 4, and Treatment 6, which were supplemented with algal and fungal oils compared to other test diets.

The contents of crude fiber and ash were 40.7 ± 7.6 (28.9–48.8) and 114 ± 4.8 (103–118 g/kg DM), respectively. Amino acid and fatty acid profiles in the six test diets fed to the *Pangasius* catfish broodstock were quite similar among the experimental diets (Table 1 and Table 3). Table 3 shows that the total contents of fatty acid profiles (g/kg DM) were highest and lowest in Treatment 6 and Treatment 2, respectively.

### 3.2. Growth Performance Indices of Broodstock

The broodstock fish from each treatment were measured to obtain the final fish body indices at the end of the experiment. The growth performance indices of the experimental fishes are shown in Table 4. The average final body weight (BW) and weight gain (WG) of female broodstock fish at the end of the experiments were 6.11 ± 0.0.46 (5.57–6.95) kg/fish and 1.24 ± 0.48 (0.63–1.95) kg/fish (wet weight basis), respectively, with slight differences between treatments (*p* < 0.05). There were significant differences (*p* < 0.05) in daily weight gain (DWG) and specific growth rate (SGR) between the different treatments (Table 4). The highest WG, DWG, and SGR were found in Treatment 5 (diets supplemented with 0.6% vitamin premix H-OVN and 1.26% algal oil), followed by Treatment 1 (diets supplemented with 0.6% vitamin premix C-2020), and the lowest were found in Treatment 4 (diets supplemented with 0.6% vitamin premix H-OVN) (Table 4).

### 3.3. Reproductive Breeding, Hatching, and Early Life Stage Development

The average diameters of the egg before and after fertilization were 955 ± 101.8 (925–985) µm and 1078.3 ± 77.8 (1023.3–1133.3) µm, respectively (*p* = 0.001–0.007). The egg sizes after fertilization were 1.1–1.2 times larger than the sizes before fertilization (Table 4). The total number of eggs in the female ovary (egg) was 886,692 (546,383–1,227,000) and was significantly different between treatments (*p* < 0.05).

The average gonad weight of females was highest in Treatment 5 (754 g/fish) and Treatment 1 (750 g/fish), followed by Treatment 6 (500 g/fish), Treatment 4 (466.7 g/fish), Treatment 3 (400 g/fish), and Treatment 2 (366.7 g/fish) (*p* < 0.05). Gonadosomatic index (GSI%) values were found highest and lowest in Treatment 5 and Treatment 3 (6.0%), respectively (*p* < 0.05) (Table 4). The relative fecundity index (RFI) was approximately 144,763 ± 77,427 (90,014–199,512) egg/kg and significantly different between treatments (*p* < 0.05). Egg fertilization and hatching ratios varied between treatments, with a range of 62.2–85.9% and 43.5–87.3%, respectively (*p* = 0.001–0.027, Figure 2). The highest proportion of egg fertilization was recorded for Treatment T5 (85.9%), followed by Treatment 6 (85.1%) and Treatment 1 (82.4%), and the highest hatching rate was in Treatment 1 (87.3%) followed in descending order by Treatment 5 (76.2%), Treatment 2 (73.7%), Treatment 6 (70.3%), Treatment 4 (54.6%), and Treatment 3 (43.5%) (Figure 2). The results regarding the feed and feeding efficiency of broodstock in each treatment showed that there was no significant difference in the food conversion ratios (FCRs) between the dietary treatments. However, we found that FCRs in treatments 3, 5, and 6 had slightly better results compared to the other treatments.

### 3.4. Growth Performance Indices of Fish Larvae 15 Days after Hatching

There were significant differences (*p* = 0.001–0.033) between treatments in final body weight gain (BWG), final length, daily weight gain (DWG), and specific growth rate (SGR) of *Pangasius* catfish larvae 15 days after hatching (Table 5). The highest and lowest final BW, final fish length, WG, and DWG were recorded in Treatment 3 (diets containing 0.6% vitamin premix H-OVN) and Treatment 2 (diets containing 0.6% vitamin premix L-OVN), respectively. The specific growth rates (SGR%) were highest in Treatment 2 (10.1%) and Treatment 4 (9.2%), followed by Treatment 6 (8.6%), Treatment 1 (8.5%), Treatment 3 (8.2%), and Treatment 5 (6.1%) (*p* < 0.05). Table 5 and Figure 3 show that the total number of fish larvae and final survival rate of larvae 15 days after hatching were 6209 ± 3102 (4015–8402) fish and 39.7 ± 21.3 (23.4–56.0%), respectively (*p* < 0.05). The highest number and highest survival rates of fish were found in Treatment 5 (56.0%) in the diets containing 0.6% vitamin premix H-OVN, followed by Treatment 3 (40.4%), Treatment 1 (36.0%), and Treatment 6 (32.7%) in descending order. The lowest final survival rates of fish larvae were found in Treatment 2 (29.4%) and Treatment 4 (22.7%), which contained 0.6% vitamin premix L-OVN-AO) + 1.26% algal oil (*p* < 0.05).

### 3.5. Growth Performance and Survival Rates of Fish Fry and Fingerlings 30 and 45 Days after Hatching

The body weight gain (BWG), final length, daily weight gain (DWG), and specific growth rate (SGR%) of fish fry and fingerlings at 15, 30, and 45 days after hatching are presented in Table 5. The growth performance indices showed that there were significant differences among the test diets (*p* ˂ 0.05). The fish body indices 45 days after hatching were 2.1–18.7 times higher than the body indices 30 days after hatching (Table 5). The total number and survival rate of fingerlings 45 days after hatching were 870 ± 464 (413.3–1326.3) fingerlings and 45.3 ± 17.4 (22.0–66.3%) (*p* < 0.05). The highest final survival rates of fingerlings were recorded in Treatment 3 (66.3%) and Treatment 5 (40.4%), followed by Treatment 6 (36.3%) and Treatment 1 (23.3%), while the lowest survival rates were found in Treatment 4 (22.7%) and Treatment 2 (22.0%).

### 3.6. Water Quality Monitoring

Average temperature, pH, DO, NH_3_/NH_4_^+^, and NO_2_^−^ of the water in the earthen pond where the broodstock were reared were approximately 27 ± 4 (24.8–29.8 °C), 7.7 ± 0.5 (30–8.05), 2.4 ± 1.0 (1.7–3.1 mg/L), 0.13 ± 0.5 (0.11–0.15 mg/L), and 0.45 ± 0.5 (0.35–0.58 mg/L), respectively. The water quality parameters of the fish larval and fingerling rearing tanks were 26.5 ± 1.5 (25–27 °C), 7.9 ± 0.1 (7.8−8.1) pH, 1.9 ± 1.1 (0.8−3.2) NO_2_^−^ (mg/L), and 0.05 ± 0.03 (0.02−0.10) NH_3_/NH_4_^+^ (mg/L).

## 4. Discussion

A diet rich in ingredients such as n-3 LC-PUFA fatty acids, essential amino acids, vitamin antioxidants, and prebiotic compounds has been shown to improve the broodstock survival rate of larvae, yield, and farmed fish quality [19,45,46,47]. Ref. [45] reported that maternal nutrition directly influences the quality of the larvae and fingerlings. Lipids (fats and fatty acids) from fish oils, vegetable oils, microalgae, and algal oils are essential macronutrients for the growth performances of fish, and they provide at least three key essential fatty acids (EFAs), which contain n-3 LC-PUFA, specifically DHA (22:6n-3) and EPA (20:5n-3). These substances are important for the metabolism of terrestrial animals and fish and contribute to their growth and physiological functions [17,18,21,48,49]. Several researchers have shown that diets containing highly unsaturated fatty acids (HUFAs), such as n-3 and n-6 HUFA, influence gonadal development, egg quality, fecundity, hatching, and larvae survival rates [15,21,50,51,52]. It is [51,53] reported that the dietary manipulation of n-3 and n-6 highly unsaturated fatty acids could improve levels and ratios of AA, EPA, and DHA, which were transferred to the resulting eggs with improvements in early survival and hatching success for European sea bass (*Dicentrarchus labra*) and Channel catfish (*Ictalurus punctuates*).

In recent years, the microalgal biomass market has produced about 5000 tonnes of dry matter per year and generates a turnover of approximately USD 136.25 million per year [54,55]. Ref. [17] reported that both fungi and algae are potential sources of natural antibiotics and antioxidants that would be safe to use and have no side effects. It is indicated that fungi and algae have the potential to replace fish meal and fish oil in aquaculture and ensure sustainability standards. Algae provide essential amino acids and valuable triglycerides, such as lipids, vitamins, and pigments, making them suitable as nutritional supplements in livestock feed and aquafeed formulations [17,18,19]. Several previous studies have reported that, among the lipids in algae, there are essential unsaturated fatty acids (EFAs), including arachidonic acid, eicosapentaenoic acid, and the rare γ-linolenic acid (GLA) [17,49,55].

Vitamins are organic compounds essential for supporting the normal growth and health of fish. They represent a significant cost in fish food and aquafeed production [56]. Since fish often cannot synthesize vitamins and essential amino acids, they must be supplied in their diets [15,16,56]. Vitamin premixes commonly used in food fish production diets are often considered adequate for farmed fish [56]. Consequently, the relative importance of fatty acid and vitamin premix contents for reproductive performance and fish fry production can be qualitatively assessed by testing them as supplements in broodstock fish diets. The chemical composition and the content of essential amino acids and fatty acids of the experimental diets used in this experiment were comparable with the values reported for broodstock diets of European Sea Bass [51], Gilthead sea bream [16], and Channel catfish [57]. Also, the contents of crude protein and crude fat and the composition of the vitamin premix compounds of the test diets in this study were in good agreement with feed given to adult channel catfish, tilapia, African catfish, and striped catfish [29,40,58].

Our results show that there were significant differences (*p* < 0.05) among the test diets in growth performance indices (final body weight, total weight gain, daily weight gain, specific growth rate) and reproduction indices (gonad somatic index, relative fecundity index, total number of eggs in the ovary, percentage of fertilized eggs, and hatching ratios) of the broodstock (Table 4 and Figure 2). The values for the diets in Treatment 5 were the highest, followed by Treatment 1, Treatment 6, Treatment 2, Treatment 3, and Treatment 4 in descending order. This indicates that the *Pangasius* catfish broodstock received the nutrients in the diet well without compromising growth and reproductive performance indices. Refs. [15,20] reported that gonadal development, fecundity and egg fertilization, egg size, and total egg volume all increased in some fish species when certain dietary proteins and essential nutrients, such as essential amino acids, vitamins, and fatty acids, were available in the feed.

The broodstock SGR in this study was similar to values observed in a study on striped catfish breeders after six months of feeding with a trial feed containing 35% crude protein and a supplementary vitamin premix [59,60]. The GSI (%) and RFI (egg/kg) values in this study were much higher than the GIS (4.73–9.21%) and RFI (65,000–168,900 eggs/kg) previously reported for *Pangasius* catfish broodstock fed different dietary proteins (25–45% CP) [59,61]. In general, the relative fecundity values (egg/kg) in this study were comparable to values (117,000–153,000 eggs/kg) of striped catfish broodstock spawners obtained by [4,59,62]. However, it was much greater than the values found for Basa fish (*Pangasius bocourti*) and *Pangasius* catfish broodstock reported by [63,64]. The fertilized egg incubation period in this experiment lasted for 33–36 h to complete the hatching process and is similar to the hatching periods reported for Asian *Pangasius* catfish [65]. The average hatching rate of eggs in the present study after the main spawning season was approximately 78.5% (*p* < 0.05), which was in the range of 70–80% hatching rates reported for *Pangasius* sutchi during peak season in Bangladesh [66], Vietnam [4,5], Malaysia [67] and Nepal [62]. However, the hatching rate of this study was much higher than the average values of 55–65% reported for Tra catfish (*Pangasius hypophthalmus*) and Basa catfish (*Pangasius bocourti*) in Vietnam [63,65,68] and 30% reported for yellowtail catfish (*Pangasius pangasius*) in Bangladesh [68].

The survival rates of fingerlings reared for 45 days after hatching were approximately from 22.0% to 66.3%. The highest final survival rate was recorded in Treatment 3 (66.3%), followed by Treatment 5 (45.3%), Treatment 6 (36.3%), Treatment 1 (23.3%), Treatment 4 (22.7%), and Treatment 2 (22.0%) (*p* < 0.05) (Table 5, Figure 3). These results suggest that diets for *Pangasius* broodstock supplemented with 0.6% H-OVN, as well as those supplemented with 0.6% H-OVN mixed with 12.6% algal oil, can improve growth performance, reproduction of the broodstock, and survival rates of fingerlings. This improvement is likely due to the adequate provision of lipids, essential amino acids, and vitamins, which regulate metabolism and intestinal flora. The final survival rates (22.0–66.3%) of fingerlings in the present study after the main spawning season were 1.4 to 2.0 times higher than the survival rates (20–25%) during the main spawning season and (12–15%) after the main spawning season reported for *Pangasius* catfish by fish farmers in the Mekong Delta [2,4,5,6,7]. This study’s results indicated that the supplementation of vitamins/vitamin premixes and plant oils (algae and fungi oils) has significantly improved the reproduction of broodstock, growth performance, and survival rates of fry and fingerlings of the *Pangasius* catfish species. However, the results of the present research need further extensive study at the fish farms and industry farms for practical application, as many scientists have indicated and reported that many factors affect the growth performance, broodstock reproduction, hatchability, and survival rates of fry and fingerlings of fish. In the early stages of fry’s lives, internal influences from the parents, such as the initial dietary quality and nutrition of the broodstock, play a key role in their development and survival rates. In the later life stages, from larvae and fry to fingerlings and adult fish, external factors such as live feeds (Moina, Artemia, Algae), diet quality, feed nutritional quality, environmental conditions, disease prevention and treatment, and fish cultivation practices are more influential in determining the growth, quality, survival rates, and overall health of the fish [35,68,69,70,71].

## 5. Conclusions

The results of this study demonstrated that the broodstock of *Pangasius* catfish effectively absorbed the nutrients from the test diets, resulting in increased growth and improved reproductive performance indices. Our research found that diets containing 35% crude protein (CP), supplemented with 0.6% vitamin premix H-OVN (Treatment 3), and diets containing 35% CP, supplemented with 0.6% vitamin premix H-OVN and 1.26% algal oil (Treatment 5), produced the best results. These diets led to optimal growth, enhanced reproductive performance, and the highest final survival rates of fish fry and fingerlings. The overall conclusions of this study indicated that the results were consistent with the hypothesis and have effectively addressed the main objectives that were expected. The findings of this research provide valuable information for *Pangasius* catfish farmers and the fish production industries in the Mekong Delta, Vietnam. Further studies on these diets for *Pangasius* catfish broodstock, along with a cost–benefit analysis under commercial farm conditions, are recommended to confirm the findings of this study.

## Figures and Tables

**Figure 1 animals-14-02203-f001:**
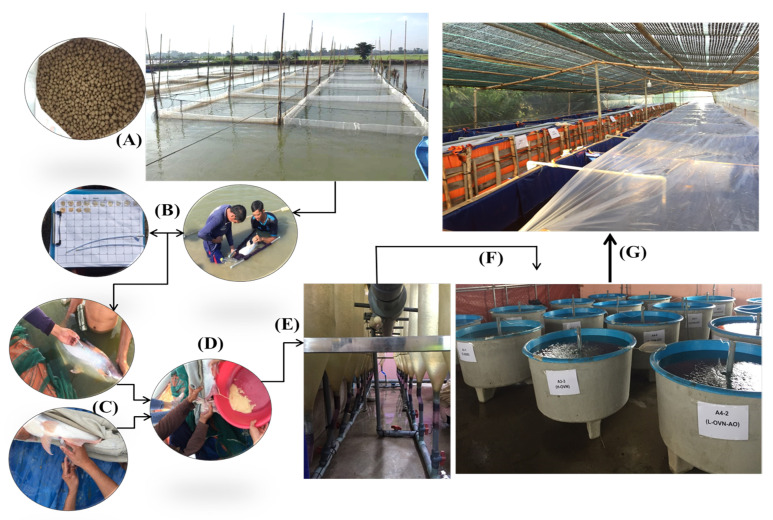
The layout of experimental broodstock rearing, reproductive performance, and larvae rearing of *Pangasius catfish*. (**A**) Experiment of broodstock fish rearing and feeding; (**B**) Broodstock selection for induced prawning; (**C**) Broodstock injection and sperm quality checking; (**D**) Egg stripping and fertilizing; (**E**) Fertilized egg incubating in the hatching incubator system; (**F**) Larvae after hatching for two days of nursing; (**G**) Experiment of larvae and fingerlings reared for 45 days after hatching.

**Figure 2 animals-14-02203-f002:**
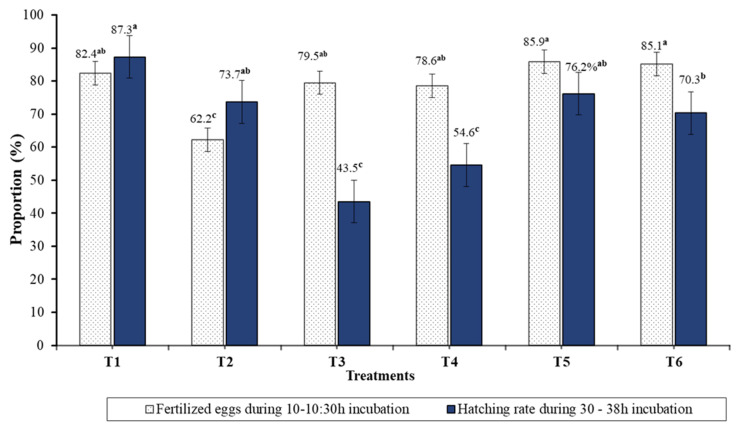
The proportion (%) of fertilized eggs and hatching rates of *Pangasius* catfish. The means within columns with different superscript letters are significantly different (*p* ˂ 0.05).

**Figure 3 animals-14-02203-f003:**
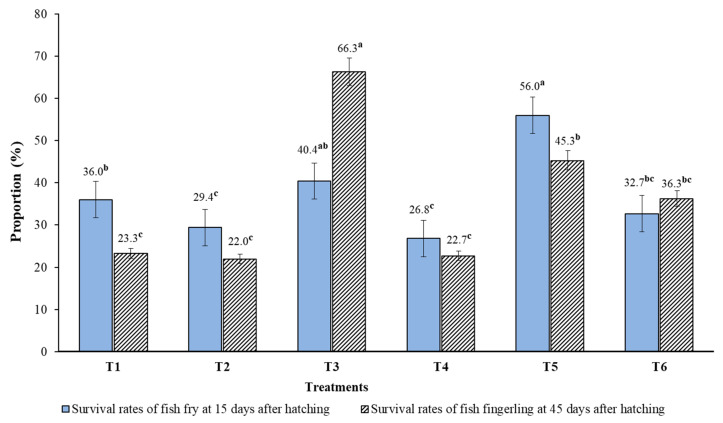
The survival rate of fish fry 15 days and fish fingerlings 45 days after hatching of *Pangasius* catfish. The means within columns with different superscript letters are significantly different (*p* ˂ 0.05).

**Table 1 animals-14-02203-t001:** Formulation, chemical composition, and amino acid profiles of test diets (g/kg DM) used for broodstock of *Pangasius* catfish species.

Raw Materials	Experimental Diets					
	T1	T2	T3	T4	T5	T6
Fish meal (578 g/kg CP)	75	75	75	75	75	75
Poultry by-product meal (648 g/kg CP)	184.1	184.1	184.1	184.1	184.1	184.1
Wheat flour (158 g/kg CP)	288.1	288.1	288.1	289.2	289.2	288
Soybean meal (490 g/kg CP)	360	360	360	360	360	360
Soybean oil	38.9	38.9	38.9	47.6	47.6	47.2
Fish oil	40	40	40	17.6	17.6	17.6
Choline chloride	5	5	5	5	5	5
Mineral premix ^a^	3	3	3	3	3	3
Vitamin premix (Rovimix 2020) ^b^	6	−	−	−	−	−
Vitamin premix (L-OVN) ^c^	−	6	−	6	−	−
Vitamin premix (H-OVN) ^d^	−	−	6	−	6	6
Algal oil ^e^	−	−	−	12.6	12.6	12.6
Fungal oil	−	−	−	−	−	1.5
Actual chemical composition (g/kg DM)						
Dry matter	901	904.1	885.2	897	895.3	930.2
Crude protein	350.1	350.5	350	351.5	358.5	351.5
Crude fat	65.2	66.8	65.2	68.4	73.8	69.8
Crude fiber	28.9	34.4	48.8	46.4	40.9	44.5
Ash	103	110	109	110	110	118
Amino acid profiles (g/kg DM)						
Histidine	6.4	6.4	6.4	6.4	6.4	6.4
Isoleucine	12.2	12.2	12.2	12.2	12.2	12.2
Leucine	23.8	23.8	23.8	23.8	23.8	23.8
Lysine	17.2	17.2	17.2	17.3	17.3	17.3
Methionine	4.6	4.6	4.6	4.6	4.6	4.6
Phenylalanine	7.7	7.7	7.7	7.7	7.8	7.8
Valine	16.9	16.9	16.9	17	17	17
Threonine	12.1	12.1	12.1	12.1	12.1	12.1
Aspartic acid	30.4	30.4	30.4	30.4	30.4	30.4
Glutamic acid	58.2	58.2	58.2	58.3	58.3	58.3
Alanine	29.4	29.4	29.4	29.4	29.4	29.4
Glycine	34.9	34.9	34.9	35	35	35
Proline	30.6	30.6	30.6	30.7	30.7	30.7
Serine	16.6	16.6	16.6	16.6	16.6	16.6
Tyrosine	7.8	7.8	7.8	7.8	7.8	7.8

Note: ^a^ Mineral premix contained (g kg^−1^): copper sulphate pentahydrate (13.33), iron sulphate monohydrate (64.52), manganese oxide (22.22), zinc oxide (62.50), cobalt (0.66), iodine (33.33), selenium (2.96), and filler (800.48). ^b^ Composition of vitamin Rovimix C-2020 contained (g kg^−1^) vitamin A (0.42), vitamin E (20.0), vitamin C (35.71), vitamin K (1.94), vitamin B1 (0.45), vitamin B2 (1.25), vitamin B6 (1.02), vitamin B12 (0.25), niacin (4.0), pantothenic acid (3.33), folic acid (0.46), biotin (0.83), choline (166.67), anticaking (10.0), BHT (0.2), and alpha cellulose (753.46). ^c^ Vitamin premix L-OVN contained (g kg^−1^) vitamin A (1.33), vitamin E (83.33), vitamin C (238.09), vitamin K (1.94), vitamin B1 (1.81), vitamin B2 (3.13), vitamin B6 (3.05), vitamin B12 (0.40), niacin (13.33), pantothenic acid (7.41), folic acid (0.83), biotin (0.83), choline (166.67), anticaking (10.0), BHT (0.2), and alpha cellulose (467.64). ^d^ Vitamin premix H-OVN contained (g kg^−1^) vitamin A (1.83), vitamin E (166.67), vitamin C (476.19), vitamin K (1.94), vitamin B1 (3.62), vitamin B2 (3.13), vitamin B6 (5.08), vitamin B12 (0.40), niacin (13.33), pantothenic acid (7.41), folic acid (2.08), biotin (0.83), choline (166.67), anticaking (10.0), BHT (0.2), and alpha cellulose (140.62). ^e^ Algal oil contained EPA + DHA: EPA-content (110 mg g^−1^), DHA + EPA content (505 mg g^−1^), TotOx (2 × Peroxide value + anisidine value: (6.0)), free fatty acid (3.6%), and moisture (0.33%). All products of vitamin solution for *Pangasius* Broodstock, algal oil, and fungal oil were provided by DSM SINGAPORE INDUSTRIAL PTE. LTD, (Company Registration No. 199100649D), a company incorporated in Singapore and having its registered office at 30 Pasir Panjang Road, #13–31, Mapletree Business City, Singapore, 117440, trading as DSM NUTRITIONAL PRODUCTS ASIA PACIFIC (Business Registration No 53192777).

**Table 2 animals-14-02203-t002:** Proximate chemical composition (g/kg DM) of feed ingredients.

	Feed Ingredients			
	Soybean Meal	Wheat Flour	Fish Meal	Poultry By-Product Meal
Dried matters	895	888	911	953
CP	490	158	578	648
Lipid	12.0	13.0	70.0	72.0
Ash	58.0	15.0	185	259
Crude fiber	26.0	4.0	4.0	26.0
Total	175.8	37.1	145.1	108.7

**Table 3 animals-14-02203-t003:** Fatty acid profiles (g/kg DM) in test diets used for broodstock of *Pangasius* catfish.

Treatments	Crude Fat	MA (C14:0)	PA (C16:0)	SA (C18:0)	OLA (C18:1n9)	LA (C18:2n6)	αNA (C18:3n3)	ARA (C20:4n6)	EPA (C20:5n3)	DHA (C22:6n3)
T1	3.01	0.043	0.556	0.393	0.820	0.597	0.056	0.010	0.015	0.053
T2	2.59	0.032	0.518	0.237	0.879	0.560	0.038	0.016	0.010	0.010
T3	3.18	0.040	0.622	0.302	1.065	0.661	0.047	0.010	0.010	0.024
T4	3.30	0.042	0.648	0.306	1.075	0.665	0.048	0.011	0.016	0.066
T5	3.01	0.038	0.590	0.261	0.953	0.641	0.047	0.013	0.024	0.082
T6	4.41	0.047	0.817	0.389	1.395	1.016	0.076	0.018	0.033	0.121

Note: T1 (Treatment 1): Rovimix 2020, 0.6% (C-2020); T2 (Treatment 2): L-OVN for *Pangasius* Broodstock 0.6% (L-OVN); T3 (Treatment 3): H-OVN for *Pangasius* Broodstock 0.6% (H-OVN); T4 (Treatment 4): L-OVN for *Pangasius* Broodstock 0.6% + 12.6% algal oil; T5 (Treatment 5): H-OVN for *Pangasius* Broodstock 0.6% + 12.6% algal oil (H-OVN-AO); T6 (Treatment 6): H-OVN for *Pangasius* Broodstock 0.6% + 12.6% algal oil + 1.5% fungal oil (H-OVN-AO/FVO).

**Table 4 animals-14-02203-t004:** Growth and reproductive performance indices of *Pangasius* catfish broodstock fed with test diets for 65 days.

Indices		Experimental Treatments						*p*-Value
		T1	T2	T3	T4	T5	T6	
Growth performance indices of brooder fish								
Initial body weight (kg)		4.4 ± 0.7	5.0 ± 0.9	5.2 ± 0.9	4.9 ± 0.8	5.0 ± 0.1	4.7 ± 0.1	0.841
Final body weight (kg)		6.2 ± 0.9 ^b^	6.1 ± 0.7 ^b^	6.0 ± 0.0 ^b^	5.6 ± 0.6 ^c^	7.0 ± 1.1 ^a^	6.0 ± 0.2	0.024
Weight gain (kg)		1.8 ± 0.2 ^ab^	1.0 ± 0.3 ^b^	0.9 ± 0.9 ^bc^	0.6 ± 0.3 ^c^	2.0 ± 0.9 ^a^	1.3 ± 0.2 ^ab^	0.027
Daily weight gain (kg)		0.04 ± 0.0 ^ab^	0.02 ± 0.01 ^b^	0.02 ± 0.02 ^bc^	0.02 ± 0.01 ^c^	0.10 ± 0.02 ^a^	0.02 ± 0.01 ^ab^	0.027
Specific growth rate (SGR%)		0.5 ± 0.4 ^ab^	0.3 ± 0.2 ^b^	0.3 ± 0.2 ^bc^	0.2 ± 0.1 ^c^	0.6 ± 0.2 ^a^	0.4 ± 0.0 ^ab^	0.027
Food conversion ratio (FCR)		1.8 ± 0.3	1.8 ± 0.4	1.7 ± 0.9	1.8 ± 0.2	1.6 ± 0.1	1.7 ± 0.3	0.100
Reproductive performance indices								
Egg size (µm)	Before injecting hCG	1.0 ± 0.01	0.9 ± 0.01	0.9 ± 0.01	1.0 ± 0.01	1.0 ± 0.01	0.9 ± 0.1	0.071
	After injecting hCG	1.1 ± 0.01	1.1 ± 0.01	1.0 ± 0.03	1.0 ± 0.05	1.1 ± 0.03	1.0 ± 0.01	0.862
Gonad weight (g/fish)		750.1 ± 54.8 ^a^	366.7 ± 100.1 ^b^	400.0 ± 109.5 ^ab^	466.7 ± 264.6 ^ab^	754.02 ± 248.1 ^a^	500.1 ± 100.6 ^ab^	0.004
Gonad somatic index (GSI%)		12.2 ± 1.0 ^a^	6.0 ± 1.7 ^b^	6.7 ± 1.8 ^ab^	8.4 ± 4.8 ^ab^	12.8 ± 0.8 ^a^	8.4 ± 0.1 ^ab^	0.027
Relative fecundity index (egg/kg)		152,158 ± 7467 ^a^	90,014 ± 2467 ^b^	104,267 ± 7381 ^ab^	133,642 ± 1503 ^a^	199,512 ± 7467 ^a^	119,748 ± 1166 ^ab^	0.022
Total number of eggs in female ovary (egg)		1,227,000 ± 107.4 ^a^	546,383 ± 303.7 ^b^	625,600 ± 452.3 ^ab^	744,387 ± 119.9 ^ab^	1,057,500 ± 232.1 ^a^	712,500 ± 106.9 ^ab^	0.047

Note: The values represent the mean ± SD (Standard Deviation). Means within rows with different superscript letters are significantly different (*p* ˂ 0.05).

**Table 5 animals-14-02203-t005:** Growth performance indices and fingerling production of *Pangasius* catfish fingerlings reared for 15 days after hatching, 30 days after hatching, and 45 days after hatching.

Indices	Experimental Treatments					
	T1	T2	T3	T4	T5	T6
Body indices of fish fry reared for 15 days (*n* = 15,000 larvae/tank)						
BWG (mg)	12.1 ± 0.1 ^c^	11.11 ± 0.0 ^d^	15.6 ± 0.1 ^a^	14.4 ± 0.2 ^b^	14.3 ± 0.2 ^b^	12.9 ± 0.2 ^c^
Length (mm)	14.9 ± 0.4 ^bc^	14.82 ± 0.4 ^c^	15.4 ± 0.5 ^a^	15.2 ± 0.4 ^ab^	15.1 ± 0.4 ^bc^	15.0 ± 0.4 ^bc^
DWG (mg)	0.6 ± 0.01 ^d^	0.60 ± 0.01 ^e^	0.8 ± 0.01 ^a^	0.8 ± 0.01 ^b^	0.8 ± 0.01 ^b^	0.6 ± 0.0 ^cd^
SGR%	8.5 ± 0.6 ^c^	10.05 ± 0.7 ^a^	8.2 ± 0.4 ^c^	9.2 ± 0.9 ^b^	9.1 ± 0.6 ^b^	8.6 ± 1.0 ^c^
Body indices of fingerlings reared for 30 days (*n* = 2000 fingerlings/tank)						
BWG (mg)	63.0 ± 35.2 ^b^	147.0 ± 24.5 ^a^	41.0 ± 9.4 ^d^	23.0 ± 15.2 ^e^	47.0 ± 11.3 ^c^	39.0 ± 18.7 ^d^
Length (mm)	25.0 ± 2.2 ^a^	22.0 ± 2.0 ^ab^	18.3 ± 0.5 ^b^	23.6 ± 0.6 ^a^	18.0 ± 0.4 ^b^	24.0 ± 1.4 ^a^
DWG (mg/d)	4.9 ± 0.3 ^b^	11.3 ± 1.5 ^a^	3.2 ± 0.8 ^d^	1.7 ± 0.9 ^e^	3.6 ± 1.5 ^c^	3.0 ± 19 ^d^
SGR (%)	6.9 ± 1.2 ^c^	10.6 ± 2.2 ^a^	4.4 ± 1.6 ^e^	2.6 ± 1.7 ^f^	6.4 ± 3.0 ^d^	4.7 ± 2.1 ^e^
Body indices of fingerlings reared for 45 days (*n* = 2000 fingerlings/tank)						
BWG (mg)	947.0 ± 64.0 ^a^	313.0 ± 58.7 ^d^	708.0 ± 53.2 ^b^	431.0 ± 66.4 ^c^	304.0 ± 51.2 ^d^	499.0 ± 98.8 ^c^
Length (mm)	60.1 ± 1.7 ^a^	41.5 ± 1.2 ^b^	41.5 ± 0.9 ^b^	56.5 ± 1.0 ^a^	37.0 ± 0.7 ^b^	57.5 ± 1.3 ^a^
DWG (mg/d)	79.0 ± 1.5 ^a^	26.0 ± 2.3 ^d^	59.0 ± 3.3 ^b^	36.0 ± 2.9 ^c^	25.0 ± 3.2 ^d^	42.0 ± 5.4 ^c^
SGR (%)	6.8 ± 0.7 ^b^	2.7 ± 0.6 ^d^	9.0 ± 1.5 ^a^	4.0 ± 2.2 ^c^	4.12 ± 2.2 ^c^	4.0 ± 1.8 ^c^
Total number of surviving fingerlings	466.7 ± 12.7 ^b^	439.3 ± 94.7 ^b^	1326.3 ± 560.5 ^a^	413.3 ± 242.7 ^b^	781.3 ± 173.2 ^b^	603.7 ± 251.7 ^b^

Note: The values represent the mean ± SD (Standard Deviation). Means within rows with different superscript letters are significantly different (*p* ˂ 0.05).

## Data Availability

Data are available from the corresponding author upon request.

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
