# Peer review of "Vitamin Solutions Effects on Reproduction of Broodstock, Growth Performance, and Survival Rate of Pangasius Catfish Fingerlings"

_animals, 2024, doi:10.3390/ani14152203_

Round 1

Reviewer 1 Report

Comments and Suggestions for Authors

The authors did an excellent job of providing relevant information on the importance of the study. The study is good and relevant and will help meet the aquaculture industry's need to improve Pangasius catfish production with fewer brooders; my suggestions are below. 

Introduction

Please provide the specific objectives of the study. These set the pace for your methodology.

Materials and Methods 

Thanks for providing the graphic of the experimental design. It helps in understanding the study.

What was the average initial weight of the broodfish used in the study? Please include the information in this section. 

Were the fish acclimatized before the start of the experiment? Information needs to be included with the number of days.

     Lines 192-193: how often was the fish fed during the acclimatization period?

Line 187: what were the factors in the factorial design? These factors must be stated in the methodology.

Lines 204-205 (Table 1), is the Actual chemical composition calculated or analyzed? Please correct this on the table.

In lines 228-229, what was the time interval for administering the four hCG injections for the female broodstock?

Results:

Table 3, with the analyzed amino acids and fatty acids compositions; the table contains no fatty acid values, but only eight essential amino acids. Why did you pick these eight amino acids?

In addition, the authors should explain how the essential amino acids end up with the same values in all the diets. Please provide adequate information about how you formulated your diets. 

How do the algae and fungi oils supplemented in some of the diets (Table 1) reflect on the analyzed compositions? The authors should include information on whether the analysis was conducted or not. 

Discussion:

Lines 439 – 453 focus on the importance of fatty acids in broodstock diets, but the present study did not determine or provide information on fatty acid compositions. Authors should include how this information relates to this study. Also, Lines 473 – 475 are also on the fatty acids.

The authors should clarify if the amino acids were supplemented in the diets and if the final results merit the practice in aquaculture. The discussion should also be improved to reflect these findings.

The authors should improve the discussion by focusing on the data provided. 

Thanks

Comments on the Quality of English Language

The authors should correct the manuscript for grammatical errors, examples are line 498 (spawners obtained by) and line 503 (off? breeding season).

Author Response

Dear Editorial office

Journal of Animals, MDPI Publisher

We are writing to submit a revised version of the paper entitled “Vitamin solutions effects on reproduction of broodstock, growth performance and survival rate of Pangasius catfish fingerling” to Journal of Animals (Manuscript ID: animals-3081777), in the section of aquatic animal and special issues: Reproduction and Development in Fish: Solving Bottlenecks in Modern Aquaculture. We are grateful to the editors and reviewers for the thoughtful comments and feedback that have improved the presentation and clarity of the paper manuscript. We have revised the text throughout in response to each of the comments and suggestions from the editors and reviewers (Please kindly see revised paper manuscript in the attached file). Please kindly find ours detailed response to the Editor and Reviewers comments below and the corresponding revisions/corrections highlighted in the Revised paper manuscript.

Point-by-point response to Comments and Suggestions for Authors

I. Introduction section

#Comments 1: Please provide the specific objectives of the study. These set the pace for your methodology

Author's Response 1: The specific objectives of this study were aim to assess the reproductive success of broodstock, growth performance, survival rates of Pangasius catfish fed by different enriched diets supplemented with vitamin solutions and plant oils. To provide practical recommendations for the Pangasius catfish farmers and aquaculture industry based on the findings of the study.

II. Materials and Methods

#Comments 2: Thanks for providing the graphic of the experimental design. It helps in understanding the study.

Author's Response 2: Many thanks!

#Comments 3: What was the average initial weight of the broodfish used in the study? Please include the information in this section.

Author's Response 3: The average initial weight of the broodfish used in the study was about 6.0±0.7 (4.5 - 7.7) kg for female and 4.5 ±0.6 (3.5 - 6.2) kg for males (It has been added into the section 2.5 of broodstock experiment design, please kindly see page 4 of 19).

#Comments 4: Were the fish acclimatized before the start of the experiment? Information needs to be included with the number of days.

Author's Response 4: Yes, the brooder fish were acclimatized with environmental condition about 15-day before the start of the experiment.

#Comments 5: Lines 192-193: how often was the fish fed during the acclimatization period?

Author's Response 5: Thank you very much for this inquiry! the brooder fish of each treatment fed by hand with two times per day at 9.00 h AM and 16.00h PM during their acclimatization period.

#Comments 6: Line 187: what were the factors in the factorial design? These factors must be stated in the methodology.

Author's Response 6: There was only one factor of diets used for broodstock in this study.

#Comments 7: Lines 204-205 (Table 1), is the Actual chemical composition calculated or analyzed? Please correct this on the table.

Author's Response 7: the chemical composition of test diets and feed ingredients was analyzed at the Advanced Laboratory, Department of Science, CTU, Vietnam (Please kindly see detailed information of chemical analysis in section 2.0, pages 7 of 18.

#Comments 8: In lines 228-229, what was the time interval for administering the four hCG injections for the female broodstock?

Author's Response 8: Thank you very much for this interesting inquiry! Times interval for administering the four hCG injections for the female broodstock were: Dose 1 (24h), Dose 2 (24h), Dose 3 (24h), and Dose 4 (only 8h).

III.  Results section

#Comments 9: Table 3, with the analyzed amino acids and fatty acids compositions; the table contains no fatty acid values, but only eight essential amino acids. Why did you pick these eight amino acids?

Author's Response 9: Thank you very much!. All analysis results of amino acids (both essential amino-acid and non – essential amino acids) have been provided in Table 3. The fatty acids compositions have also been provided in Table 4.

#Comments 10: In addition, the authors should explain how the essential amino acids end up with the same values in all the diets. Please provide adequate information about how you formulated your diets.

Author's Response 10: Thank you for your comments and recommendations; It based the same proportions of initial feed ingredients in diets formulations of each treatment. Therefore, analysis results of amino acid contents of each treatment are same values in all the diets.

#Comments 11: How do the algae and fungi oils supplemented in some of the diets (Table 1) reflect on the analyzed compositions? The authors should include information on whether the analysis was conducted or not.

Author's Response 11: The algae oil, fungi oil, and fish oil in each diet formulations were not analysis, we only relied on detailed information that is commercially available on the market that it provided by company. The algae oil, fungi oil, and fish oil in each diet/each treatment added into the oil spraying system of a small aquafeed production plant. We did not mix the oils manually or by hand because it would not be uniform.

IV. Discussion section

#Comments 12: Lines 439 – 453 focus on the importance of fatty acids in broodstock diets, but the present study did not determine or provide information on fatty acid compositions. Authors should include how this information relates to this study. Also, Lines 473 – 475 are also on the fatty acids.

Author's Response 12: We totally agree with your comments and suggestions (sorry for missed this data). The contents of fatty acids profiles in broodstock diets have been provided in the paper manuscript (Please kindly see Table 4, page 9 of 19 in paper manuscript). Thank you very much!

#Comments 13: The authors should clarify if the amino acids were supplemented in the diets and if the final results merit the practice in aquaculture. The discussion should also be improved to reflect these findings.

Author's Response 13: Thank you very much! We would like to confirm that the amino acids were not supplemented in test diets in this study. We analyzed the amino acids profiles of test diets with the main aim of determining whether did the diets for each treatment in this experimented provide nutritionally adequate for the fish or not?

#Comments 14: The authors should improve the discussion by focusing on the data provided.

Author's Response 14: Thank you very for your suggestions! the discussion section has been revised and improved.

4. Response to Comments on the Quality of English Language

Author's Response: The English has been edited by our colleague from University of Stirling, UK.

5. Additional clarifications: No thanks

Thank you very much for your time to review our paper!

Reviewer 2 Report

Comments and Suggestions for Authors

Manuscript entitled "Vitamin solutions effects on reproduction of broodstock, growth performance and survival rate of Pangasius catfish fingerling" for consideration in Animals. The manuscript presents interesting and useful findings on the effects of vitamin solution supplementation on Pangasius catfish broodstock and fingerling production. However, some areas need further clarification and interpretation, particularly regarding the statistical analyses and the interpretation of the results in relation to the hypothesis tested. Additionally, the language and style could benefit from proofreading and editing for clarity and coherence. Here are my comments and suggestions for improvements. Therefore I recommend its major revision.

·       Revise the abstract to highlight the main findings and implications of the study more clearly. Specifically, mention the hypothesis tested, the statistical methods used, and the main conclusions drawn from the data.

·       Make sure to include the exact values of the statistical tests used to determine significance values

·       Use past tense instead of future tense when describing the results i.e.will provide" should be changed to provided.

·       Provide more background information on the importance of Pangasius catfish in Vietnamese aquaculture and the challenges faced by farmers in producing high-quality fingerlings.

·       state the hypothesis tested in the study and how it relates to previous research.

·       Organize the introduction in a logical manner, starting with broader context and narrowing down to the specific focus of the study.

·       Ensure that all necessary details are provided for replication purposes, such as the exact dosages and compositions of the vitamin solutions used, the duration of the experiment, and the sample sizes.

·       Describe the statistical analyses performed in detail, including the type of ANOVA used, the pairwise comparison method, and the software used.

·       Clarify the units used for measurements (e.g., grams vs. kilograms) and make sure they are consistent throughout the text.

·       Present the data in a clear and concise manner, using tables and figures where appropriate.

·       Use past tense when describing the results (e.g., "showed" instead of "show").

·       Interpret the results in light of the hypothesis tested and previous literature.

·       Provide actual numbers rather than just ranges whenever possible i.e. report the exact percentages rather than saying "ranging between...".

·       Begin by summarizing the main findings of the study and relating them to the original hypothesis.

·       Compare and contrast the results with previous research, highlighting strengths and limitations of both.

·       Explore the mechanisms underlying the observed effects, drawing on existing literature and theory.

·       Address potential weaknesses or limitations of the study and propose directions for future research.

·       Summarize the main contributions and implications of the study.

·       Emphasize the practical relevance of the findings for farmers and the aquaculture industry.

·       Highlight the novelty and significance of the study in the broader context of aquaculture research.

Comments on the Quality of English Language

A through revision is required

Author Response

Dear Editorial office

Journal of Animals, MDPI Publisher

We are writing to submit a revised version of the paper entitled “Vitamin solutions effects on reproduction of broodstock, growth performance and survival rate of Pangasius catfish fingerling” to Journal of Animals (Manuscript ID: animals-3081777), in the section of aquatic animal and special issues: Reproduction and Development in Fish: Solving Bottlenecks in Modern Aquaculture. We are grateful to the editors and reviewers for the thoughtful comments and feedback that have improved the presentation and clarity of the paper manuscript. We have revised the text throughout in response to each of the comments and suggestions from the editors and reviewers (Please kindly see revised paper manuscript in the attached file). Please kindly find ours detailed response to the Editor and Reviewers comments below and the corresponding revisions/corrections highlighted in the Revised paper manuscript.

#Comments 1: Revise the abstract to highlight the main findings and implications of the study more clearly. Specifically, mention the hypothesis tested, the statistical methods used, and the main conclusions drawn from the data.

Authors' response 1: Thank you very much your comment and suggestion for the abstract section. The abstract has been revised as your suggested and recommended (Please kindly see the red text in the abstract).

#Comments 2: Make sure to include the exact values of the statistical tests used to determine significance values.Response 2: Thank you for this suggestion, we would like to confirm that the values of the statistical tests are checked and corrected.

Authors' response 2: Thank you for this suggestion, we would like to confirm that the values of the statistical tests are checked and corrected.

#Comments 3: Use past tense instead of future tense when describing the results i.e. will provide" should be changed to provided.

Authors' response 3: Many thanks, it has been corrected and edited in the text of paper manuscript!

#Comments 4: Provide more background information on the importance of Pangasius catfish in Vietnamese aquaculture and the challenges faced by farmers in producing high-quality fingerlings.

Authors' response 4: Thank you! Some background information and the challenges faced by farmers have been provided in the section of introduction (Please kindly see page 2 of 19).

#Comments 5:  Organize the introduction in a logical manner, starting with broader context and narrowing down to the specific focus of the study.

Authors' response 5: Thank you for your recommendation and suggestions! The organizing of the introduction and the statements of research in the introduction section have been revised and improved (Please kindly see the revised introduction section, page 2-3 of 19).

#Comments 6: Ensure that all necessary details are provided for replication purposes, such as the exact dosages and compositions of the vitamin solutions used, the duration of the experiment, and the sample sizes.

Authors' response 6: Yes, we would like to confirm that all detailed information of the exact dosages and compositions of the vitamin solutions used, the duration of the experiment, and the sample sizes are presented in the experimental design, feed formulation, induced prawning of broodstock (please kindly see the section of Materials and Methods).

#Comments 7: Describe the statistical analyses performed in detail, including the type of ANOVA used, the pairwise comparison method, and the software used.

Authors' response 7: We have provided the detailed statistical analyses performed for this study in section 2.11. Statistical analysis, page 8 of 19.

#Comments 8: Clarify the units used for measurements (e.g., grams vs. kilograms) and make sure they are consistent throughout the text.

Authors' response 8: Many thanks, the units used for measurements in the research article have been checked.

#Comments 9: Present the data in a clear and concise manner, using tables and figures where appropriate.

Authors' response 9: Thank you so much for your suggestion! We will check and correct carefully.

#Comments 10: Use past tense when describing the results (e.g., "showed" instead of "show").

Authors' response 10: Many thanks, we have checked and corrected it.

#Comments 11: Interpret the results in light of the hypothesis tested and previous literature.

Authors' response 11: Yes!, we have already checked and interpreted the results in light of the hypothesis tested in the study (please kindly see the hypothesis tested of our research in the introduction section).

#Comments 12: Provide actual numbers rather than just ranges whenever possible i.e. report the exact percentages rather than saying "ranging between..."

Authors' response 12: Yes, we have revised and provide all actual numbers in the text of the paper manuscript.

#Comments 13: Begin by summarizing the main findings of the study and relating them to the original hypothesis.

Authors' response 3: Yes, we have already provided a short sentence of the original hypothesis in the section of conclusion.

#Comments 14: Compare and contrast the results with previous research, highlighting strengths and limitations of both.

Authors' response 14: Thank you so much for this interesting suggestion and recommendation! In the process of reviewing the literature and identifying the problem issues, and statement problem that need to be addressed. I would like to confirm that in Vietnam, as well as in some Southeast Asian countries, there are very few studies on the feed nutrition of broodstock of Pangasius catfish similar to our study. Most of the previously studies primarily focused only on feed and feeding for the nursery and grow-out farms. The limitation of this research is that we did yet not consider about the cost-benefit analysis. Therefore, we would recommend that further studies on these diets for Pangasius catfish broodstock, along with a cost-benefit analysis under commercial farm conditions, are recommended to confirm the findings of this study.

#Comments 15: Explore the mechanisms underlying the observed effects, drawing on existing literature and theory.

Authors' response 15: Yes, I would like to confirm that it is clear that vitamin solutions/vitamin premixes and plant oils (algae oil, fungal oils) can play multifaceted roles in enhancing the reproductive performance of broodstock, promoting the growth of fingerlings, and improving their survival rates. The observed effects of vitamin supplementation in the study are consistent with the known physiological and biochemical functions of these essential nutrients. By ensuring supplement of vitamin solutions and plant oils in enrich-diets, fish farmers can improve the health and productivity of their farms.

#Comments 16: Address potential weaknesses or limitations of the study and propose directions for future research.

Authors' response 16: Thank you, in order to address potential weaknesses or limitations of this study further study should more consider to study on the effects of feed nutrition on the health improvement, disease resistance of both broodstock and fingerling, and along with a cost-benefit under the commercial farm conditions.

#Comments 17: Summarize the main contributions and implications of the study.

Authors' response 17: The study "Vitamin solutions effects on reproduction of broodstock, growth performance and survival rate of Pangasius catfish fingerling" makes significant contributions to understanding the role of vitamins and plant oils supplementation in broodstock fish nutrition. Its implications for aquaculture practices, nutritional guidelines, sustainability, and future research are substantial, providing a foundation for improving the health, productivity, quality and survival rates of fingerling of Pangasius catfish farming.

#Comments 18: Emphasize the practical relevance of the findings for farmers and the aquaculture industry.

Authors' response 18: Yes, the practical relevance of the findings from this study on vitamin solutions/vitamin premixes and plant oils effects are profound. For fish farmers, it means better reproductive success, improved growth performance, increased survival rates, higher net-incomes, and more sustainable farming practices. For the aquaculture industry, it promotes the adoption of high-quality feeds, provides enough a good quality fish fingerlings year-round, encourages sustainable practices, and supports the development of industry standards, ultimately leading to a more efficient and profitable sector.

#Comments 19: Highlight the novelty and significance of the study in the broader context of aquaculture research.

Authors' response 19: Yes, By highlighting the critical role of vitamins and plan oils supplement in fish health and development, the research underscores the importance of balanced nutrition in achieving sustainable and profitable aquaculture practices. The study's novelty lies in its focus on broodstock nutrition and its detailed analysis of multiple vitamins and plant oils' effects. Its significance is evident in the practical benefits for fish farmers and the aquaculture industry, including improved reproductive success, growth performance, survival rates, economic gains, and sustainability. The research's contributions enrich the broader field of aquaculture and pave the way for further advancements in fish nutrition and health.

Thank you very much for your time to review our paper!

Round 2

Reviewer 1 Report

Comments and Suggestions for Authors i. Table 1 shows the composition of the diets without any information about the amino acids that were supplemented.

ii. In my first review, I requested the authors to provide more information on the diets and clarify any errors. The author responded that the diets were formulated to meet the requirements of the fish, which further complicated the matter.

iii. Only a few fish have the amino acid requirements established in fish nutrition. For instance, scientists can only conveniently balance the first few limiting amino acids (essential amino acids) in rainbow trout. The nonessential amino acids are important but are not essential (not included in the diet). Hence, when supplemented diets are analyzed in the lab after formulation, the amino acids profile of the diets will reflect the formulation. In most cases,  they are not accurate. 

iv. Table 3 presented in this study cannot be accurate because the study did not supplement any essential amino acids in the diets (Table 1). Therefore,  how can the values of the amino acids in all the diets be exactly the same when they were analyzed? Further, if any essential amino acids were supplemented in the diets, how can the values for the nonessential amino acids be the same for all the diets?

Please check the following articles.   https://doi.org/10.1046/j.1365-2109.1997.t01-1-00836.x https://doi.org/10.1016/j.fsi.2020.01.026 https://doi.org/10.1016/j.anifeedsci.2022.115428 DOI:10.26609/avas.642  https://doi.org/10.1016/j.anifeedsci.2020.114593

Author Response

The comments and suggestions from Reviewer

#Table 1 shows the composition of the diets without any information about the amino acids that were supplemented.

#In my first review, I requested the authors to provide more information on the diets and clarify any errors. The author responded that the diets were formulated to meet the requirements of the fish, which further complicated the matter.

# Only a few fish have the amino acid requirements established in fish nutrition. For instance, scientists can only conveniently balance the first few limiting amino acids (essential amino acids) in rainbow trout. The nonessential amino acids are important but are not essential (not included in the diet). Hence, when supplemented diets are analyzed in the lab after formulation, the amino acids profile of the diets will reflect the formulation. In most cases,  they are not accurate.

#Table 3 presented in this study cannot be accurate because the study did not supplement any essential amino acids in the diets (Table 1). Therefore, how can the values of the amino acids in all the diets be exactly the same when they were analyzed? Further, if any essential amino acids were supplemented in the diets, how can the values for the nonessential amino acids be the same for all the diets?

Authors’ response to Reviwer 1 :

First, On behalf of authors of the paper manuscript with titled “Vitamin solutions effects on reproduction of broodstock, growth performance and survival rate of Pangasius catfish fingerling” (Paper manuscript ID: animals-3081777). I would like to extend our sincerely thanks to you for taking your time to provide the review and suggestions our paper manuscript, helping us to improve both quality and content of this study in a clearer manner.

We have taken your feedback seriously and have responded the following specific revisions: We have provided the amino acid composition of test diets in Table 1 (Please see Table 1, pages 5-6). Table 3, which contained the amino acid composition, has been removed to avoid the confusion and complexity for future readers. Additionally, the amino acid composition supplementation is not a main objective of our study (thank you very much for this valuable suggestion).

We totally agree with your points that there are currently not many fish species with their amino acid requirements identified and estimated, which is particularly true for Pangasius broodstock as there is no existing research. We have also referred to the published references you provided, and we found that these studies supplement certain amino acids for fish/shrimp as you mentioned. Once again, we sincerely appreciate and acknowledge your valuable feedback and review of our research.

Thank you very much for your time to review our paper!

Reviewer 2 Report

Comments and Suggestions for Authors

Article can be accepted now.

Comments on the Quality of English Language

a through revision for language in terms of phrasing is need 

Author Response

Dear Professor!

On behalf of authors of the paper manuscript, I would like to scenerly thanks for your time to review our paper!

Have a nice day and best regards,

Thi Da